# Distributed Adaptive Formation Control with Collision Avoidance and Connectivity Maintenance of Multiple Autonomous Surface Vehicles

1st Quan Shi
*State Key Laboratory of*
*Ocean Engineering*
*Shanghai Jiao Tong University*
Shanghai, China
quanshi@sjtu.edu.cn

2nd Xin Li
*State Key Laboratory of*
*Ocean Engineering*
*Shanghai Jiao Tong University*
Shanghai, China
lixin@sjtu.edu.cn

3rd Jianmin Yang
*State Key Laboratory of*
*Ocean Engineering*
*Shanghai Jiao Tong University*
Shanghai, China
jmyang@sjtu.edu.cn

*Abstract*—In this paper, a distributed adaptive formation control with collision avoidance and connectivity maintenance strategy is proposed for multiple autonomous surface vehicles (ASVs) subject to unknown uncertainties and disturbances. Four control objectives of formation generation, formation maintenance, collision avoidance, and connectivity maintenance can be achieved simultaneously by integrating the artificial potential field (APF) methods into the leader-follower strategies. While the two stages of distributed formation control, namely formation generation and formation maintenance, can be achieved through information exchange among inter-vehicles. The APF method provides auxiliary repulsive and attractive force to assist the ASVs in achieving collision avoidance and connectivity maintenance among inter-vehicles. Furthermore, each vehicle faces unknown dynamics due to model uncertainty and environmental disturbances, which increases the complexity of the system and hampers the achievement of control objectives. To solve these problems, neural network (NN) technologies are employed and their learning parameters are designed in scalar form. Only one scalar learning parameter instead of the tremendous weight matrix of NN needs to be adaptively updated for each vehicle. In this way, the computational burden can be greatly reduced. According to Lyapunov stability theory and graph theory, the proposed controller can be proved to accomplish the four control objectives. Several sets of comparative simulations verify the effectiveness of the proposed controller.

*Index Terms*—Formation Control, Collision Avoidance, Connectivity Maintenance, Autonomous Surface Vehicles, Artificial Potential Field, Neural-Network.

## I. INTRODUCTION

Cooperative formation control of multiple agents, such as autonomous vehicles or mobile vehicles, has received widespread attention in system and control engineering [1]–[5]. In the field of marine engineering, the cooperative formation of ASV group also has extensive applications, such as exploration of marine resources, environmental monitoring and rescue operations, etc. It contributes to improving work effi-

This work is supported by the Major Science and Technology Projects of Hainan Province (No. ZDKJ2019001) and Ministry of Industry and Information Technology of China (MC-202030-H04), to which the authors are most grateful.

ciency and promoting automation. Generally speaking, the formation control strategies can be mainly included into leader-follower strategies [6]–[9], virtual structure strategies [10] and behavior-based strategies [11]. Many control technologies are employed to solve the cooperative control problems of multiple ASVs, such as sliding mode control, backstepping-based algorithms, dynamic surface control, etc. Cooperative formation control usually divides into two stages, namely formation generation and formation maintenance. However, the limitations of the above results are twofold. On the one hand, the problems of collision avoidance among multiple ASVs were not fully considered in [6]–[11]. Collisions among inter-vehicles in multi-vehicle systems can prevent vehicles from generating the desired formation pattern [12], [13]. On the other hand, the problems of connectivity maintenance were not considered in [6]–[13]. In cooperative formation control, the communication range usually limits the connectivity of multiple ASVs. Therefore, the excessive separation distance induced by the collision avoidance behaviors will have a negative impact on connectivity maintenance among inter-vehicles [14], [15]. Consequently, the cooperative formation control of multiple ASVs inevitably faces the problems of collision avoidance and connectivity maintenance.

The previous works on the control methods for collision avoidance and connectivity maintenance can be mainly divided into three categories, which are desired formation switching methods [16], prescribed performance control (PPC) methods [17]–[20], and artificial potential field (APF) methods [21]–[25]. A PPC-based decentralized formation control with collision avoidance and connectivity maintenance of ASVs was proposed in [19]. The dynamic surface control (DSC) technologies were introduced into the kinematic design to avoid the use of unmeasured accelerations of vehicles. In [20], an observer-based decentralized adaptive output feedback formation control of USVs was designed under collision avoidance and connectivity maintenance constraints. The formation control with collision avoidance and connectivity maintenance of multiple ships was proposed based on novel APF methods

in [22]. In [23], an elliptical approximation of the ships was adopted to replace the previous circle approximation, thereby improving the operability and performance of collision avoidance and connection maintenance. In [24], an observer-based cooperative formation control with collision avoidance and connectivity maintenance of ASVs was developed by integrating the velocity potential functions into the kinematics guidance laws. The nonlinear tracking differentiators (NTD) were utilized to avoid the use of unmeasured accelerations of vehicles and to simplify the complexity of the proposed control laws without relying on DSC technologies.

Although the previous works have achieved many meaningful results, the formation control of ASVs is still an open issue. It is worth pointing out that most of the above results were concentrated on the centralized and decentralized formation architectures [18]–[23]. However, the centralized and decentralized formation architectures are poor at exchanging information among inter-vehicles, which are difficult to perform collision avoidance and maintain connectivity. In distributed architectures, ASVs can implement control decisions by exchanging information about itself and its topological neighbors. Communication topology, with respect to information exchange, plays an important role in distributed cooperative formation control [26]. The well-developed graph theory can provide sufficient theoretical support for the communication topology among ASVs. The distributed formation architectures are more suitable for performing collision avoidance and connectivity maintenance by exchanging information among inter-vehicles. The distributed formation architectures also have strong robustness and extensibility. Note that we adopt the classification views from [18]–[20] and [32], where the distributed architectures do not refer to a special kind of one-to-one communication topology. It is also worth noting that the above backstepping-based cooperative formation control of ASVs passively uses the unmeasured acceleration state of ASVs due to the virtual control law or its derivative forms in previous results [21]–[23]. Although some measures have been integrated into the backstepping-based algorithms to solve the above problems, these measures have also increased the complexity of the controller, such as DSC technologies [17]–[20] and NTD technologies [24], [25]. Consequently, from a practical perspective, it is necessary to study a simplified distributed formation control with collision avoidance and connectivity maintenance of ASVs.

Additionally, the kinetics of ASV are characterized as nonlinearity, strong coupling and subject to model uncertainties and environment disturbances. The cooperative motion control of ASVs is a durable research topic. Particularly, unknown uncertainties and external disturbances also reduce the cooperative control performance of the system. NN technologies are usually utilized as approximations in the nonlinear systems because of its universal approximation and learning ability [17]–[20]. In the previous literatures, the ideal approximation accuracy can be obtained at the cost of increasing the number of neural network nodes and learning parameters. If such NN-based algorithms are applied to the cooperative

motion control engineering of actual ASVs, it will increase the computational burden and running cost. Fortunately, some control theories and applications have been reported to solve the aforementioned computational burden problems [27], [28]. In [27], the sliding mode formation controllers were studied for multiple marine vessels with unknown nonlinear hydrodynamics. In order to reduce the learning parameters, the NN was utilized to approximate the unknown nonlinear hydrodynamic collectively. The adaptive learning parameters of the NN are designed in scalar form to reduce the computational burden of nonlinear systems in [28]. Although the results of literature [28] only focus on consensus control, it can be flexibly extended to cooperative formation control of ASVs through some approaches.

Motivated by above observations, this paper study the distributed adaptive formation control of ASVs subject to unknown uncertainties and disturbances. Through information exchange, follower-vehicles can track the position and orientation of the leader-vehicle to achieve the desired formation pattern. The APF methods are employed to assist the formation control of ASVs with the capability of collision avoidance and connectivity maintenance. Specifically, the unknown uncertainties and disturbances are recognized as unknown kinetic entireties and approximated by NN. The scalar forms of adaptive weight parameters are design to approximate unknown dynamics and greatly reduce the number of learning parameters. It can be proved that the proposed distributed adaptive formation controller can realize the four control objectives of formation generation, formation maintenance, collision avoidance and connectivity maintenance for ASVs subject to unknown uncertainties and disturbances. The hydrodynamic parameters of each vehicle are set to be different to verify the effectiveness of the proposed distributed controller to unknown dynamics. Three cases of comparative simulations are designed to verify the superiority of the proposed controller for multiple ASVs formation control with collision avoidance and connectivity maintenance.

Compared with previous work, the contributions of this article are reflected as follows: Firstly, compared to the centralized and decentralized formation control in [18]–[23], this paper proposes a distributed formation control algorithm for ASVs. Moreover, the proposed algorithm can simultaneously realize the four control objectives of formation generation, formation maintenance, collision avoidance and connectivity maintenance by integrating the APF methods into the leader-follower strategies. Secondly, this paper proposed a distributed formation control algorithm for multiple ASVs without relying on DSC methods or NTD methods, which not only avoids the use of unmeasured acceleration, but also reduces the complexity of the controller. Thirdly, compared with previous NN-based formation control of ASVs [17]–[20], [25] and [27], the number of learning parameters of the proposed controller can be greatly reduced by designing learning parameters in scalar form. Since the proposed distributed formation controller only requires one learning parameter to be adaptively updated for each vehicle, which can greatly reduce the computational

burden.

The remaining parts of this paper are organized as follows: Section II present the Preliminaries; Section III introduces problem formulation; Section IV draws the main results; Section V provides the simulation results; Section VI gives the conclusion.

## II. PRELIMINARIES

### A. Algebraic Graph Theory

In this paper, we are concerned with the formation behaviors and information exchange of ASVs through communication topology network. The ASVs systems are defined as vertices of the graph. This communication topology network is denoted as edges of the graph corresponding to the information exchange among inter-vehicles. The communication graph can be indicated as $\mathcal{G} = (\mathcal{V}, \mathcal{E}, \mathcal{A})$, where $\mathcal{V} = \{\mathcal{V}_1, \mathcal{V}_2, \dots, \mathcal{V}_n\}$, $\mathcal{E} \subseteq \mathcal{V} \times \mathcal{V}$ and $\mathcal{A} = [a_{ij}]$ represent vertex set, edge set and adjacency matrix, respectively. The edge $\mathcal{E}_{ij} = (\mathcal{V}_i, \mathcal{V}_j) \in \mathcal{E}$ indicate that the information flow is communicated from vertex $\mathcal{V}_i$ to vertex $\mathcal{V}_j$, where the vertex $\mathcal{V}_j$ can be denoted as a neighbor of vertex $\mathcal{V}_i$. The neighbor set can be indicated as $N_i = \{\mathcal{V}_j : (\mathcal{V}_i, \mathcal{V}_j) \in \mathcal{E}\}$. As regards adjacency matrix $\mathcal{A} = [a_{ij}]$, the element $a_{ij}$ denotes the edge weight corresponding to the edge $\mathcal{E}_{ij}$, where $a_{ij} > 0$ represents the information exchange between ASV $i$ and ASV $j$, otherwise $a_{ij} = 0$ and $a_{ii} = 0$. A graph can be termed as an undirected graph, if $a_{ij} = a_{ji}$. The edge weight between follower ASV $i$ and leader ASV is described as $\mathcal{B} = \text{diag}\{b_1, b_2, \dots, b_n\}$, where $b_i > 0$ represents the information exchange between follower and leader, otherwise $b_i = 0$.

Moreover, the Laplacian matrix $\mathcal{L} = [l_{ij}] \subset \mathbb{R}^{n \times n}$ of graph $\mathcal{G}$ is defined as $\mathcal{L} = \mathcal{D} - \mathcal{A}$, where $\mathcal{D} = \text{diag}\{d_1, d_2, \dots, d_n\}$, $d_i = \sum_{j=1}^{n} a_{ij}$, and $i = 1, 2, \cdots n$.

*Assumption 1:* For the distributed leader-follower formation framework, at least one follower ASV is assumed to be connected to the leader ASV.

*Lemma 1:* The undirected graph $\mathcal{G}$ is connected if and only if its Laplacian matrix $\mathcal{L}$ is irreducible.

*Lemma 2:* If the Laplacian matrix $\mathcal{L} = [l_{ij}] \subset \mathbb{R}^{n \times n}$ of the undirected graph $\mathcal{G}$ is irreducible, then the eigenvalues of the matrix $\widetilde{\mathcal{L}} = \mathcal{L} + \mathcal{B} = \begin{bmatrix} l_{11} + b_1 & \cdots & l_{1n} \\ \vdots & \ddots & \vdots \\ l_{n1} & \cdots & l_{nn} + b_n \end{bmatrix}$ are positive.

### B. ASV System Description

In the multiple ASVs systems, the 3-DOF motional model of ASV can be described as follows [29], [30]:

$$\begin{cases} \dot{\eta}_i = R_i(\psi_i) v_i \\ M_i \dot{v}_i = -C_i(v_i) v_i - D_i(v_i) v_i + \tau_i + \tau_{wi} \\ i = 1, 2, \cdots, n \end{cases} \quad (1)$$

where $\eta_i = [x_i, y_i, \psi_i]^T \in \mathbb{R}^3$ indicates the position $p_i = (x_i, y_i) \in \mathbb{R}^2$ and heading angle $\psi_i$ under the Earth-fixed frame. $v_i(t) = [u_i(t), v_i(t), r_i(t)]^T \in \mathbb{R}^3$ is the velocity vector under the Body-fixed frame, of which $u_i$, $v_i$ and $r_i$ are the surge velocity, sway velocity and yaw rate corresponding to 3-DOF, respectively. $\tau_i(t) = [\tau_{ui}, \tau_{vi}, \tau_{ri}]^T$ is the control input. $\tau_{wi} \in \mathbb{R}^3$ is the bounded and continuous external disturbances. $M_i = M_i^T \in \mathbb{R}^{3 \times 3}$, $C_i = -C_i^T \in \mathbb{R}^{3 \times 3}$ and $D_i \in \mathbb{R}^{3 \times 3}$ respectively denote the added mass effects and inertial matrix, the Coriolis force and centripetal matrix, and the hydrodynamic damping matrix.

The rotational matrix $R_i(\psi_i) = \begin{bmatrix} \cos\psi_i & -\sin\psi_i & 0 \\ \sin\psi_i & \cos\psi_i & 0 \\ 0 & 0 & 1 \end{bmatrix}$ denotes the coordinate transformation from the Body-fixed frame to the Earth-fixed frame, of which $R_i^{-1}(\psi_i) = R_i^T(\psi_i)$, $\|R_i(\psi_i)\| = 1$, $\dot{R}_i(\psi_i) = R_i(\psi_i) S_i(r)$, $R_i^T(\psi_i) S_i(r) R_i(\psi_i) = S_i(r)$ are the unique properties, where $S_i(r) = \begin{bmatrix} 0 & -r_i & 0 \\ r_i & 0 & 0 \\ 0 & 0 & 0 \end{bmatrix}$.

### C. Artificial Potential Functions and Virtual Forces

In this paper, APF method is utilized to maintain an appropriate separation distance between two ASVs. The repulsive potential function and attractive potential function are assembled to achieve such control objectives. The geometric illustration of multiple vehicles for collision avoidance and connectivity maintenance are displayed in Fig 1.

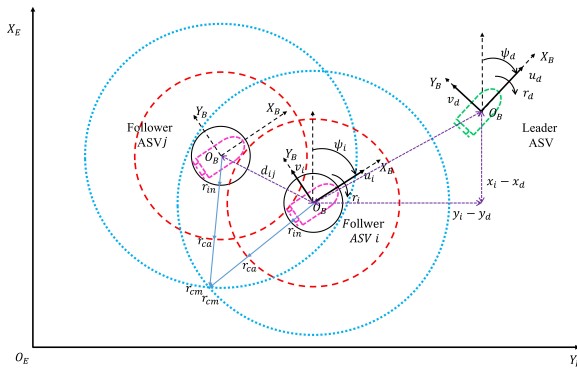

Fig. 1. Geometric illustration of multiple vehicles for collision avoidance and connectivity maintenance.

*Definition 1 ( [15], [32]):* The differentiable and nonnegative function $P_r(d_{ij}(t))$ can be defined as repulsive potential function if it holds: 1)

1) $P_r(d_{ij}(t)) \to \infty$ when $\|d_{ij}(t)\| \to r_{ca}$;
2) $P_r(d_{ij}(t)) = 0$ when $\|d_{ij}(t)\| \to d_{ca}$;
3) $P_r(d_{ij}(t))$ is strictly reduced in the range of $\|d_{ij}(t)\| \in (r_{ca}, d_{ca}]$.

where $d_{ij}(t)$ indicates the relative position variable between vehicle $i$ and $j$, $r_{ca}$ denotes the collision avoidance radius with respect to minimum safe separation distance, $d_{ca}$ expresses the collision avoidance detection radius with respect to triggering repulsive potential function.

*Definition 2 ( [15], [32]):* The differentiable and nonnegative function $P_a\left(d_{ij}\left(t\right)\right)$ can be defined as attractive potential function if it holds: 1)

1) $P_a\left(d_{ij}\left(t\right)\right) \to \infty$ when $\|d_{ij}\left(t\right)\| \to r_{cm}$;
2) $P_a\left(d_{ij}\left(t\right)\right) = 0$ when $\|d_{ij}\left(t\right)\| \to d_{cm}$;
3) $P_a\left(d_{ij}\left(t\right)\right)$ is strictly increased in the range of $\|d_{ij}\left(t\right)\| \in [d_{cm}, r_{cm}]$;

where $r_{cm}$ means the connectivity maintenance detection radius, $d_{cm}$ represents the connectivity maintenance radius with respect to maximum separation distance.

Based on the aforementioned definitions, the total potential function $P\left(d_{ij}\left(t\right)\right)$ can be obtained as follows

$$P\left(d_{ij}\left(t\right)\right) = P_r\left(d_{ij}\left(t\right)\right) + P_a\left(d_{ij}\left(t\right)\right) \tag{2}$$

The attractive and repulsive virtual forces can be calculated from the repulsive and attracted potential functions, respectively. Furthermore, the total potential field force can be derived as follows

$$\delta\left(d_{ij}\right) = \delta_r + \delta_a = -\nabla P_r\left(d_{ij}\right) - \nabla P_a\left(d_{ij}\right) \tag{3}$$

where $-\nabla$ represent the negative gradient. Once vehicle $j$ enters the collision avoidance detection range of vehicle $i$, $j$ can be called the collision avoidance neighbor of $i$, and its set can be described as $N_i^{ca} = \{j \,|\|d_{ij}\left(t\right)\| \le d_{ca}\}$. Once vehicle $j$ enters the connectivity maintenance detection range of vehicle $i$, $j$ can be called the connectivity maintenance neighbor of $i$, and its set can be described as $N_i^{cm} = \{j \,|\|d_{ij}\left(t\right)\| \ge d_{cm}\}$.

*Remark 1:* The collision avoidance neighbor $N_i^{ca}$ and connectivity maintenance neighbor $N_i^{cm}$ are different from topological neighbor $N_i$. $N_i^{ca}$ and $N_i^{cm}$ are only related to the distance between the two ASVs.

## III. PROBLEM FORMULATION

Consider a group of ASVs, of which the mathematical model of the $i$th ASV is described as equation (1). In order to overcome the difficulties caused by the coupling of system state variables to the formation control with collision avoidance of ASVs. The following coordinate transformation form is employed to transform equation (1) as the following form [40]:

$$\begin{cases} \dot{\eta}_i = \omega_i \\ \dot{\omega}_i = f_i\left(z\right) + \tau_{ci} \end{cases} \tag{4}$$

where $\omega_i = R_i\left(\psi_i\right)v_i$ and $\tau_{ci} = R_i M_i^{-1}\tau_i$ are the velocity vector and control input vector of the transformed system, respectively. $f_i\left(z\left(t\right)\right) = S_i\omega_i - R_i M_i^{-1}C_i\left(R_i^{-1}\omega_i\right)R_i^{-1}\omega_i + R_i M_i^{-1}D_i\left(R_i^{-1}\omega_i\right)R_i^{-1}\omega_i + R_i M_i^{-1}\tau_{wi}R_i^{-1}\omega_i \in R^3$ represents collective nonlinear dynamics including internal uncertainties and external disturbances, which are conceived as unknown dynamics. Note that in order to reduce adaptive learning parameters, the collective dynamics $f_i\left(\cdot\right)$ of follower-ASV are considered as unknown entireties and approximated by radial basis function neural networks (RBF-NNs).

The ASV group is guided by a virtual leader-ASV, and its mathematical model can be expressed as follows:

$$\begin{cases} \dot{\eta}_d = R_d\left(\psi_d\right)v_d \\ M_d\dot{v}_d = -C_d\left(v_d\right)v_d - D_d\left(v_d\right)v_d f_d\left(x_d, v_d\right) \end{cases} \tag{5}$$

where $\eta_d = [x_d, y_d, \psi_d]^T \in \mathbb{R}^3$, $v_d = [u_d, v_d, r_d]^T \in \mathbb{R}^3$ are position and velocity state vectors of the leader ASV, respectively. $f_d \in \mathbb{R}^3$ represents desired smooth unknown dynamics. Define $\omega_d = R_d\left(\psi_d\right)v_d$, then the dynamics of ASV $i$ and leader ASV can be rewritten as follows

$$\begin{cases} \dot{\eta}_d = \omega_d \\ \dot{\omega}_d = f_{\omega d}\left(\eta_d, \omega_d, f_d\right) \end{cases} \tag{6}$$

where the unknown nonlinear function $f_{\omega d} = S_d\omega_d - R_d M_d^{-1}C_d\left(R_d^{-1}\omega_d\right)R_d^{-1}\omega_d - R_d M_d^{-1}D_d\left(R_d^{-1}\omega_d\right)R_d^{-1}\omega_d + R_d M_d^{-1}f_d$ is bounded with an assumption of $|f_{\omega d}| < \alpha$.

Define the position and velocity tracking error dynamic as follows:

$$\begin{cases} \bar{\eta}_i = \eta_i - \eta_d - \xi_i \\ \bar{\omega}_i = \omega_i - \omega_d \end{cases} \tag{7}$$

where $\bar{\eta}_i$ is with respect to position and heading, $\bar{\omega}_i$ is with respect to velocity and yaw, $\xi_i = [\xi_{ix}, \xi_{iy}, \xi_{i\psi}]^T \in \mathbb{R}^3$ denotes the desired relative position vector between the ASV $i$ and the leader ASV.

The control objective: A distributed adaptive formation control with collision avoidance and connectivity maintenance strategy is designed for multiple ASVs. The desired formation pattern can sail along the prescribed shape. Meanwhile, multiple ASVs can achieve collision avoidance and connectivity maintenance performance in the process of vehicles formation pattern forming and sailing by combining APF method with the leader-follower formation framework.

**(1) Formation Generation:** The ASVs are driven to achieve a desired formation pattern with relative position and orientation satisfying $\lim_{t\to\infty}\|\bar{\eta}_i\left(t\right)\| = 0$

**(2) Formation maintenance:** The velocity of the ASVs must be consistent to maintain formation while performing formation tracking objectives to satisfy $\lim_{t\to\infty}\|\bar{\omega}_i\left(t\right)\| = 0$.

**(3) Collision avoidance:** The collision among vehicles can be avoided if satisfied $\|d_{ij}\left(t\right)\| > r_{ca}$ at any time, Otherwise, the multiple ASVs fail to avoid collision.

**(4) Connectivity maintenance:** The connectivity among vehicles can be maintained if satisfied $\|d_{ij}\left(t\right)\| < r_{cm}$ at any time, Otherwise, the connectivity of multiple ASVs will be failed.

*Remark 2:* In fact, such a coordinate transformation (4) transforms the controller from the Body-fixed frame to the Earth-fixed frame. Note that the controller designed later is for dynamic (4). In order to achieve the original motional model (1), the control law should be left multiplied by matrix $M_i R_i^T$.

## IV. CONTROLLER DESIGN

Using the aforementioned dynamics (4) and (5), the following error dynamic can be obtained:

$$\begin{cases} \dot{\eta}_i(t) = \omega_i(t) - \omega_d(t) \\ \dot{\omega}_i(t) = f_i(z(t)) + \tau_{ci}(t) - f_{\omega d}(t) \end{cases} \quad (8)$$

The error dynamic (8) can be rewritten in following vector form as

$$\dot{\bar{E}} = -(P \otimes I_3)\bar{E} + \begin{bmatrix} 0_{nm} \\ F \end{bmatrix} + \begin{bmatrix} 0_{nm} \\ U \end{bmatrix} - \begin{bmatrix} 0_{nm} \\ F_{\omega d} \end{bmatrix} \quad (9)$$

where $\bar{E} = [\bar{\eta}^T, \bar{\omega}^T]^T$, $\bar{\eta} = [\bar{\eta}_1^T, \cdots, \bar{\eta}_n^T]^T$, $\bar{\omega} = [\bar{\omega}_1^T, \cdots, \bar{\omega}_n^T]^T$, $F^T = [f_1^T(z), \cdots f_n^T(z)]^T$, $F_{\omega d}^T = [f_{wd}^T, \cdots f_{wd}^T]^T$, $U = [\tau_{c1}^T, \cdots, \tau_{cn}^T]^T$, $P = \begin{bmatrix} 0_n & -I_n \\ 0_n & 0_n \end{bmatrix}$. $\otimes$ represents the Kronecker product. Because the nonlinear function $f_i(z)$ is unknown, it cannot be directly applied in controller design.

In order to obtain available controllers, RBF-NNs are employed to approximate unknown nonlinear dynamics as following forms [31]:

$$f_i(z) = W_i^{*T} H_i(z_i) + \varepsilon_i \quad (10)$$

The formation tracking error associated with position and velocity can be obtained as

$$\begin{cases} e_i^\eta = \sum_{j \in N_i} a_{ij}((\eta_i - \eta_j) - (\xi_i - \xi_j)) + b_i(\eta_i - \eta_d - \xi_i) \\ e_i^\omega = \sum_{j \in N_i} a_{ij}(\omega_i - \omega_j) + b_i(\omega_i - \omega_d) \\ i = 1, 2, \cdots, n \end{cases} \quad (11)$$

Using the aforementioned tracking error dynamic (7), the formation tracking error (11) can be rewritten as follows:

$$\begin{cases} e_i^\eta = \sum_{j \in N_i} a_{ij}(\bar{\eta}_i - \bar{\eta}_j) + b_i\bar{\eta}_i \\ e_i^\omega = \sum_{j \in N_i} a_{ij}(\bar{\omega}_i - \bar{\omega}_j) + b_i\bar{\omega}_i \\ i = 1, 2, \cdots, n \end{cases} \quad (12)$$

The distributed adaptive formation controller can be designed as

$$\tau_{ci} = -K_{1i}(e_i^\eta + e_i^\omega) - K_{2i}\hat{w}_i^T \|H_i(z_i)\|^2 (e_i^\eta + e_i^\omega) + K_{3i}\delta_i \quad (13)$$

where $K_{1i}$, $K_{2i}$ and $K_{3i}$ are positive gain parameters, $\hat{w}_i$ is estimation of $w_i^*$, and $w_i^* = \|W_i^*\|_F^2$.

The adaptive laws of estimation weight parameters $\hat{w}_i$ can be designed as follows:

$$\dot{\hat{w}}_i = \Upsilon_i\left(K_{2i}\|H_i(z_i)\|^2\|e_i^\eta + e_i^\omega\|^2 - \sigma_i\hat{w}_i\right) \quad (14)$$

where $\Upsilon_i$ and $\sigma_i$ are positive parameters.

*Remark 3:* The adaptive NN weight parameters designed in scalar form are extended for current working conditions to approximate nonlinear dynamics, which is the estimation of

the norm of the optimal weight matrix [28]. The number of adaptive weight parameters of the proposed algorithm can be greatly reduced to only one for each vehicle compared to other previous NN-based formation control algorithms [17]–[20], [25], [27].

**Theorem 1:** Consider that multiple ASVs system (1) with bounded initial state under undirected communication topology. If all the assumptions are satisfied, then the proposed distributed adaptive formation controller (13) can achieve the four control objectives, i.e., all error signals are semi-globally uniformly ultimately bounded (SGUUB) and all vehicles can maintain an appropriate distance to avoid collisions and maintain connectivity through the assistance of the potential fields.

The proof of Theorem 1 is divided into two parts, Part 1 proves the formation control behavior, and Part 2 proves the collision avoidance and connectivity maintenance behavior.

*Part A:* Define the Lyapunov function as follows:

$$V_L(t) = \frac{1}{2}\bar{E}^T(t)(Q \otimes I_3)\bar{E}(t) + \sum_{i=1}^{n} \Upsilon_i^{-1}\tilde{w}_i^T(t)\tilde{w}_i(t) \quad (15)$$

where $Q = \begin{bmatrix} 2\widetilde{\mathcal{L}} & \widetilde{\mathcal{L}} \\ \widetilde{\mathcal{L}} & \widetilde{\mathcal{L}} \end{bmatrix}$, $\widetilde{\mathcal{L}} = \mathcal{L} + \mathcal{B}$. According to Lemma 2, $\widetilde{\mathcal{L}}$ is a positive definite matrix, the Lyapunov function $V$ is also a positive definite function.

Substituting (9) into the time derivative of , we have

$$\dot{V}_L = \frac{1}{2}\dot{\bar{E}}^T(Q \otimes I_3)\bar{E} + \frac{1}{2}\bar{E}^T(Q \otimes I_3)\dot{\bar{E}} + \sum_{i=1}^{n} \Upsilon_i^{-1}\tilde{w}_i^T\dot{\hat{w}}_i$$

$$= -\frac{1}{2}\bar{E}^T\left((P^T \otimes I_3^T)(Q \otimes I_3) + (Q \otimes I_3)(P \otimes I_3)\right)\bar{E}$$

$$+ \bar{E}^T(Q \otimes I_3)\begin{bmatrix} 0_{nm} \\ F + U - F_{\omega d} \end{bmatrix} + \sum_{i=1}^{n} \Upsilon_i^{-1}\tilde{w}_i^T\dot{\hat{w}}_i$$

$$= -\frac{1}{2}\bar{E}^T\left((P^TQ + QP) \otimes I_3\right)\bar{E} + \sum_{i=1}^{n} \Upsilon_i^{-1}\tilde{w}_i^T\dot{\hat{w}}_i$$

$$+ \bar{E}^T(Q \otimes I_3)\begin{bmatrix} 0_{nm} \\ F + U - F_{\omega d} \end{bmatrix} \quad (16)$$

Using the following fact of

$$P^TQ + QP = \begin{bmatrix} 0_n & -\widetilde{\mathcal{L}} \\ -\widetilde{\mathcal{L}} & -\widetilde{\mathcal{L}} \end{bmatrix} \quad (17)$$

$$\bar{E}^T(Q \otimes I_3)\begin{bmatrix} 0_{nm} \\ F + U - F_{\omega d} \end{bmatrix} = \sum_{i=1}^{n}(e_i^\eta + e_i^\omega)^T(f_i + \tau_i^c - f_{wd}) \quad (18)$$

Equation (16) can be rewritten as

$$\dot{V}_L = -\frac{1}{2}\bar{E}^T\left(\begin{bmatrix} 0_n & -\widetilde{\mathcal{L}} \\ -\widetilde{\mathcal{L}} & -\widetilde{\mathcal{L}} \end{bmatrix} \otimes I_3\right)\bar{E} + \sum_{i=1}^{n}(e_i^\eta + e_i^\omega)^T(f_i + \tau_i^c - f_{wd}) + \sum_{i=1}^{n} \Upsilon_i^{-1}\tilde{w}_i^T\dot{\hat{w}}_i \quad (19)$$

Substituting (10) into (19), we have

$$\dot{V}_L = -\bar{E}^T \left( \begin{bmatrix} 0_n & -\widetilde{\mathcal{L}} \\ -\widetilde{\mathcal{L}} & -\widetilde{\mathcal{L}} \end{bmatrix} \otimes I_3 \right) \bar{E}$$
$$+ \sum_{i=1}^{n} (e_i^\eta + e_i^\omega)^T \left( W_i^{*T} H_i(z_i) + \varepsilon_i \right) \tag{20}$$
$$+ \sum_{i=1}^{n} (e_i^\eta + e_i^\omega)^T (\tau_i^c - f_{wd}) + \sum_{i=1}^{n} \Upsilon_i^{-1} \tilde{w}_i^T \dot{\hat{w}}_i$$

Based on Young's inequality and Cauchy inequality [28], the following results can be obtained as

$$\sum_{i=1}^{n} (e_i^\eta + e_i^\omega)^T W_i^{*T} H_i(z_i)$$
$$\leqslant \sum_{i=1}^{n} K_{2i} \left( (e_i^\eta + e_i^\omega) \left( W_i^{*T} H_i(z_i) \right) \right)^2 + \frac{1}{4K_{2i}} \tag{21}$$
$$\leqslant \sum_{i=1}^{n} K_{2i} w_i^* \|H_i(z_i)\|^2 \|e_i^\eta + e_i^\omega\|^2 + \frac{1}{4K_{2i}}$$

$$\sum_{i=1}^{n} -(e_i^\eta + e_i^\omega)^T f_l(x_l, v_l) \leqslant \sum_{i=1}^{n} \frac{1}{2} \|e_i^\eta + e_i^\omega\|^2 + \frac{\alpha^2}{2} \tag{22}$$

$$\sum_{i=1}^{n} (e_i^\eta + e_i^\omega)^T \varepsilon_i \leqslant \sum_{i=1}^{n} \frac{1}{2} \|e_i^\eta + e_i^\omega\|^2 + \frac{\beta^2}{2} \tag{23}$$

Using inequalities (21)-(23), the result (20) can be derived as

$$\dot{V}_L \leqslant -\bar{E}^T \left( \begin{bmatrix} 0_n & -\widetilde{\mathcal{L}} \\ -\widetilde{\mathcal{L}} & -\widetilde{\mathcal{L}} \end{bmatrix} \otimes I_3 \right) \bar{E} + \sum_{i=1}^{n} \|e_i^\eta + e_i^\omega\|^2$$
$$+ \sum_{i=1}^{n} K_{2i} w_i^* \|H_i(z_i)\|^2 \|e_i^\eta + e_i^\omega\|^2 + \sum_{i=1}^{n} \Upsilon_i^{-1} \tilde{w}_i^T \dot{\hat{w}}_i \tag{24}$$
$$+ \sum_{i=1}^{n} (e_i^\eta + e_i^\omega)^T (\tau_i^c) + \sum_{i=1}^{n} \left( \frac{1}{4K_{2i}} + \frac{\beta^2 + \alpha^2}{2} \right)$$

Substituting distributed adaptive formation controller (13), adaptive learning parameters (14) into (24), one has

$$\dot{V}_L \leqslant -\bar{E}^T \left( \begin{bmatrix} 0_n & -\widetilde{\mathcal{L}} \\ -\widetilde{\mathcal{L}} & -\widetilde{\mathcal{L}} \end{bmatrix} \otimes I_3 \right) \bar{E} - \sum_{i=1}^{n} \sigma_i \tilde{w}_i^T \hat{w}_i^T$$
$$+ \sum_{i=1}^{n} (1 - K_{1i}) \|e_i^\eta + e_i^\omega\|^2 - \sum_{i=1}^{n} K_{3i} \delta_i (e_i^\eta + e_i^\omega)^T \tag{25}$$
$$+ \sum_{i=1}^{n} \left( \frac{1}{4K_{2i}} + \frac{\beta^2}{2} + \frac{\alpha^2}{2} \right)$$

Outside the collision avoidance and connectivity maintenance range $r_{ca} < \|d_{ij}(t)\| < r_{cm}$, $\delta_i = 0$. Hence $\sum_{i=1}^{n} (e_i^x(t) + e_i^v(t))^T (-K_{3i}\delta_i) = 0$ [24]. Based on the fact of $\tilde{w}_i^T(t) \hat{w}_i^T(t) = \frac{1}{2} \left( \tilde{w}_i^2(t) + \hat{w}_i^2(t) - w_i^{*2} \right)$, and following inequality can be obtained

$$-\sigma_i \tilde{w}_i \hat{w}_i \leqslant -\frac{1}{2} \sigma_i \tilde{w}_i^2 + \frac{1}{2} \sigma_i w_i^{*2} \tag{26}$$

Substituting inequality (26) into (25), and after several manipulations, one has

$$\dot{V}_L \leqslant -\bar{E}^T \left( (K_{1i}-1) \begin{bmatrix} \widetilde{\mathcal{L}}^2 & \widetilde{\mathcal{L}}^2 \\ \widetilde{\mathcal{L}}^2 & \widetilde{\mathcal{L}}^2 \end{bmatrix} - \begin{bmatrix} 0_n & \widetilde{\mathcal{L}} \\ \widetilde{\mathcal{L}} & \widetilde{\mathcal{L}} \end{bmatrix} \otimes I_3 \right) \bar{E}$$
$$- \frac{1}{2} \sum_{i=1}^{n} \sigma_i \tilde{w}_i^2 + \sum_{i=1}^{n} \left( \frac{1}{4K_{2i}} + \frac{\beta^2 + \alpha^2 + \sigma_i w_i^{*2}}{2} \right) \tag{27}$$

Let $\vartheta_i > 0$ and satisfy the limit of $\vartheta_i \leqslant K_{1i} - 1$, the inequality can be transformed as

$$\dot{V}_L \leqslant -\bar{E}^T \left( (\vartheta_i \Theta - \Lambda) \otimes I_m \right) \bar{E} - \frac{1}{2} \sum_{i=1}^{n} \sigma_i \tilde{w}_i^2 + \Delta \tag{28}$$

where $\Theta = \begin{bmatrix} \widetilde{\mathcal{L}}^2 \widetilde{\mathcal{L}}^2 \\ \widetilde{\mathcal{L}}^2 \widetilde{\mathcal{L}}^2 \end{bmatrix}$, $\Lambda = \begin{bmatrix} 0_n \widetilde{\mathcal{L}} \\ \widetilde{\mathcal{L}} \widetilde{\mathcal{L}} \end{bmatrix}$, $\Delta = \sum_{i=1}^{n} \left( \frac{1}{4K_{2i}} + \frac{\beta^2 + \alpha^2 + \sigma_i w_i^{*2}}{2} \right)$.

Using the result of $\vartheta_i \Theta - \Lambda = \begin{bmatrix} \vartheta_i \widetilde{\mathcal{L}}^2 & \vartheta_i \widetilde{\mathcal{L}}^2 - \widetilde{\mathcal{L}} \\ \vartheta_i \widetilde{\mathcal{L}}^2 - \widetilde{\mathcal{L}} & \vartheta_i \widetilde{\mathcal{L}}^2 - \widetilde{\mathcal{L}} \end{bmatrix}$ and linear matrix inequality, $\vartheta_i \widetilde{\mathcal{L}}^2 - \widetilde{\mathcal{L}} > 0$, $\vartheta_i \widetilde{\mathcal{L}}^2 - \left( \vartheta_i \widetilde{\mathcal{L}}^2 - \widetilde{\mathcal{L}} \right) = \widetilde{\mathcal{L}} > 0$, so we get the result of matrix $\begin{bmatrix} \vartheta_i \widetilde{\mathcal{L}}^2 & \vartheta_i \widetilde{\mathcal{L}}^2 - \widetilde{\mathcal{L}} \\ \vartheta_i \widetilde{\mathcal{L}}^2 - \widetilde{\mathcal{L}} & \vartheta_i \widetilde{\mathcal{L}}^2 - \widetilde{\mathcal{L}} \end{bmatrix} > 0$. Further, setting the parameter to satisfy the limit of $\vartheta_i > \frac{1}{\lambda_{\min}^\Theta} \left( \lambda_{\max}^\Lambda + \frac{\mu}{2} \lambda_{\max}^Q \right)$, where $\lambda_{\min}^\Theta$ denoting the smallest eigenvalue of the matrix $\Theta$. $\lambda_{\max}^\Lambda$ and $\lambda_{\max}^P$ pointing the largest eigenvalue of the matrix $\Lambda$ and $Q$, respectively. In addition, $\mu = \min\{\sigma_1 \Upsilon_1, \cdots, \sigma_n \Upsilon_n\}$. Inequality (28) can be displayed as follows:

$$\dot{V}_L \leqslant -\mu \left( \frac{1}{2} \bar{E}^T (Q \otimes I_3) \bar{E} + \frac{1}{2} \sum_{i=1}^{n} \tilde{w}_i^2 \right) + \Delta$$
$$= -\mu V + \Delta \tag{29}$$

Based on [35, Lemma 1], the following inequality result can be obtained as

$$V_L(t) \leqslant V_L(0) e^{-\mu t} + \frac{\Delta}{\mu} \left( 1 - e^{-\mu t} \right) \tag{30}$$

Consequently, the distributed adaptive formation control with collision avoidance and connectivity maintenance performance can be achieved by setting appropriate parameters.

*Part B:* The collision avoidance and connectivity maintenance performance is analyzed only for vehicle $i$ and vehicle $j$, and the others can be analyzed by the same way.

Define the quadratic energy function as follows:

$$V_c(t) = \frac{1}{2} d_{ik}^T(t) d_{ik}(t) + \frac{1}{2} \omega_i^T(t) \omega_i(t) \tag{31}$$

Taking the time derivative of (31), one has

$$\begin{aligned} \dot{V}_c &= d_{ik}^T \dot{d}_{ik} + \omega_i^T \dot{\omega}_i \\ &= d_{ik}^T (\dot{\eta}_i - \dot{\eta}_k) + \omega_i^T (f_i + \tau_{ci}) \\ &= d_{ik}^T (\omega_i - \omega_k) + \omega_i^T \big( f_i - K_{1i}(e_i^\eta + e_i^\omega) \\ &\quad - K_{2i} \hat{w}_i^T \|H_i(z_i)\|^2 (e_i^\eta + e_i^\omega) + K_{3i}\delta_i \big) \end{aligned} \tag{32}$$

where $d_{ik}$ represents the relative position variable between vehicle $i$ and vehicle $k$.

Because the dwell time $t$ of each vehicle is finite in the region of $\Omega_{ik} = \{\, d_{ik}(t)|\, 0 \leqslant \|d_{ik}(t)\| \leqslant r_{cm}\}$. According to the remark and assumption, the terms $d_{ik}$, $\omega_i$, $\omega_k$, $e_i^\eta$, $e_i^\omega$, $f_i(z)$ and $\hat{w}_i^T(t)\|H_i(z_i)\|^2$ are continuous and bounded in this region. According to the design of the artificial potential function, the repulsion potential function and the attractive potential function will not work at the same time, so the two cases can be discussed separately.

When ASV $k$ enters the collision avoidance detection range of vehicle $i$ ($k \in N_i^{ca}$), the repulsive potential function can be triggered to compel the vehicles to separate. According to Definition 1, $\omega_i^T(t)K_{3i}\delta_i = -\omega_i^T(t)K_{3i}\nabla P_r(d_{ik}) \to +\infty$, if $\|d_{ik}\| \to r_{ca}$. Therefore, the following inequality can be obtained as

$$\omega_i^T \delta_i > \frac{1}{2}d_{ik}^T d_{ik} + \frac{1}{2}\omega_i^T \omega_i - \frac{1}{K_{3i}}\left(d_{ik}^T(\omega_i - \omega_k) + \omega_i^T \wp_i\right) \quad (33)$$

where $\wp_i = f_i(z) - K_{2i}\hat{w}_i^T\|H_i(z_i)\|^2(e_i^\eta + e_i^\omega) - K_{1i}(e_i^\eta + e_i^\omega)$.

Substituting inequality (33) into (32) and (31), one gets

$$\dot{V}_c > K_{3i}V_c \quad (34)$$

According to [33, Lemma 6], the following inequality can be obtained as

$$d_{ik}^T(t)d_{ik}(t) > 2e^{K_{3i}(t-t_0)}V_c(t_0) - \omega_i^T(t)\omega_i(t) \quad (35)$$

Since the continuous term $\omega_i^T(t)\omega_i(t)$ is bounded in the region of $\Omega_{ik}$. The inequality $2e^{K_{3i}(t-t_0)}V_c(t_0) - \omega_i^T(t)\omega_i(t) > (r_{ca})^2$ can be obtained by designing the positive gain parameter $K_{3i}$ to be large enough and setting appropriate initial state. Further, the main conclusion can be obtained as $\|d_{ik}(t)\| > r_{ca}$. So, the collision avoidance performance can be achieved by the proposed controller.

When vehicle $k$ enters the connectivity maintenance detection range of vehicle $i$ ($k \in N_i^{cm}$), the attractive potential function can be triggered to compel the vehicles to collect. According to Definition 2, $\omega_i^T(t)K_{3i}\delta_i = -\omega_i^T(t)K_{3i}\nabla P_a(d_{ik}) \to -\infty$, if $\|d_{ik}\| \to r_{cm}$. Therefore, the following inequality can be obtained as

$$\omega_i^T \delta_i < \frac{1}{2}d_{ik}^T d_{ik} + \frac{1}{2}\omega_i^T \omega_i - \frac{1}{K_{3i}}\left(d_{ik}^T(\omega_i - \omega_k) + \omega_i^T \wp_i\right) \quad (36)$$

Substituting inequality (36) into (32) and (31), one gets

$$\dot{V}_c < K_{3i}V_c \quad (37)$$

According to [33, Lemma 6], the following inequality can be obtained as

$$d_{ik}^T(t)d_{ik}(t) < 2e^{K_{3i}(t-t_0)}V_c(t_0) - \omega_i^T(t)\omega_i(t) \quad (38)$$

Since the continuous term $\omega_i^T(t)\omega_i(t)$ is bounded in the region of $\Omega_{ik}$. If the appropriate initial state and positive gain parameter $K_{3i} < \frac{1}{t-t_0}\ln\frac{r_{cm}^2 + \|\omega_i\|^2}{2V_c(t_0)}$ are satisfied, the inequality results can be obtained as $2e^{K_{3i}(t-t_0)}V_c(t_0) - \omega_i^T\omega_i < (r_{cm})^2$. Further, the main conclusion can be obtained as $\|d_{ik}(t)\| < r_{cm}$, i.e., the connectivity maintenance performance can be achieved by the proposed controller. As a result, by setting appropriate initial positions and parameters, all four control objectives can be achieved.

## V. SIMULATION RESULTS

To verify the effectiveness of the proposed control scheme, we assume that a fleet of ASVs include a leader-ASV and four follower-ASVs. Note that the different parameter matrices are set to four scale-down follower-ASVs to simulate unknown hydrodynamics in the following simulation examples [34], [35]. The initial state of leader ASV is given as $\eta_d(0) = \left[6, 2, \frac{\pi}{4}\right]^T$; The initial state of follower-ASVs are set as $\eta_1(0) = \left[4.5, 2, \frac{\pi}{4}\right]^T$, $\eta_2(0) = \left[4, 1, \frac{\pi}{2}\right]^T$, $\eta_3(0) = \left[7, 2, \frac{\pi}{4}\right]^T$ and $\eta_4(0) = \left[6, 1, \frac{\pi}{3}\right]^T$, respectively. The desired relative position vectors are defined as $\xi_1 = \left[\frac{3}{2}, 0, 0\right]^T$, $\xi_2 = \left[0, -\frac{3}{2}, 0\right]^T$, $\xi_3 = \left[0, \frac{3}{2}, 0\right]^T$ and $\xi_4 = \left[-\frac{3}{2}, 0, 0\right]^T$.

The edge weight between follower ASV $i$ and leader ASV is described as $\mathcal{B} = \text{diag}\{0, 0, 1, 0\}$. The Laplacian matrix $\mathcal{L}$ and adjacency matrix $\mathcal{A}$ of communication topology are defined as follows:

$$\mathcal{L} = \begin{bmatrix} 2 & -1 & 0 & -1 \\ -1 & 3 & -1 & -1 \\ 0 & -1 & 2 & -1 \\ -1 & -1 & -1 & 3 \end{bmatrix}, \mathcal{A} = \begin{bmatrix} 0 & 1 & 0 & 1 \\ 1 & 0 & 1 & 1 \\ 0 & 1 & 0 & 1 \\ 1 & 1 & 1 & 0 \end{bmatrix}.$$

The parameters of APF are defined as $r_{ca} = 1$, $d_{ca} = 1.5$, $d_{cm} = 3.5$, $r_{cm} = 4.0$ and $K_{3i} = 30$. The repulsive and attractive forces corresponding to the APF are expressed as follows:

$$\delta_r = -\nabla P_r(d_{ij}(t)) = \frac{d_{ca} - \|d_{ij}(t)\|}{(d_{ca} - r_{ca})(\|d_{ij}(t)\| - r_{ca})},$$

$$\delta_a = -\nabla P_a(d_{ij}(t)) = \frac{d_{cm} - \|d_{ij}(t)\|}{(d_{cm} - r_{cm})(\|d_{ij}(t)\| - r_{cm})}.$$

### A. Formation control of ASVs

In previous works, the hydrodynamics of multiple ASVs were simulated with the same parameters. In order to verify the effectiveness of the proposed controller against the unknown uncertainties and disturbances, the different parameter matrices are raised in this paper. The simulation results of formation control for Case A are shown in Fig.2-3. Fig. 2 shows the formation trajectories of multiple ASVs without considering collision avoidance and connectivity maintenance performance. First, all follower-vehicles can perform good formation control tasks, i.e. formation generation and formation maintenance. Fig.3 shows the distance between all follower-vehicles. At the initial moment, the distance between vehicles (ASV-2 and ASV-4) (ASV-1 and ASV-4) is less than the minimum separation distance, which represents the occurrence of multiple inter-vehicle collisions. It can be further concluded that the formation controller in Case A is unable to avoid inter-vehicle collisions without taking any measures.

### B. Formation control with collision avoidance of ASVs

The simulation results of formation control for Case B are shown in Fig.4-5. Fig.4 displays the formation trajectories of multiple ASVs with considering collision avoidance performance. Compared with the formation trajectories in Fig. 2, the generation trajectories of formation are significantly changed at the initial moment. Fig.5 illustrates the distance between all follower-vehicles. It can be seen that all distances between any

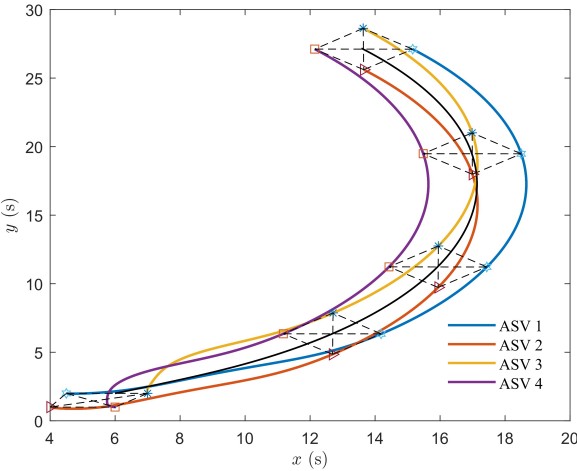

Fig. 2. The performance of formation control.

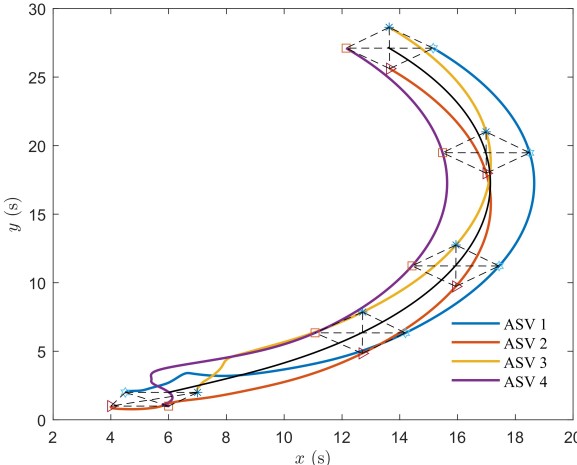

Fig. 4. The performance of formation control with collision avoidance.

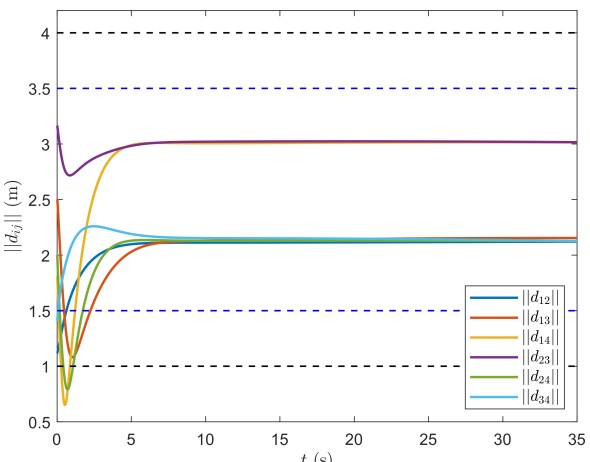

Fig. 3. The distance between all inter-vehicles.

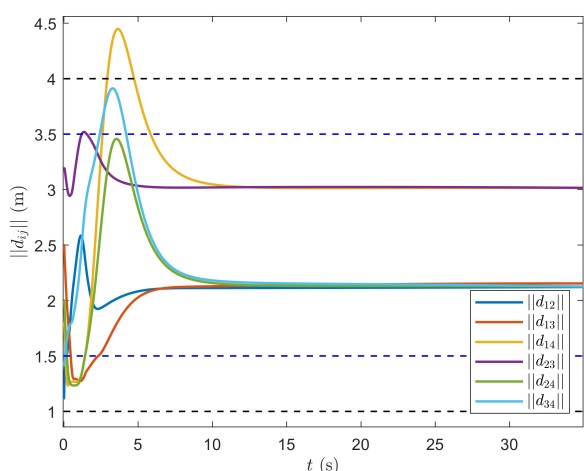

Fig. 5. The distance between all inter-vehicles.

two inter-vehicles are greater than the minimum separation distance at any time, which means that the multiple inter-vehicle collisions in Case A are successfully avoided. However, the distances are greater than the maximum separation distance corresponds to the limited sensing range of inter-vehicle. As explained in the section of introduction, collision avoidance behavior poses the problem of connectivity failure. Note that according to the simulation results of Case A and Case B, the foresight can be confirmed to be necessary for the formation control with collision avoidance and connectivity maintenance of multiple ASVs.

### C. Formation control with collision avoidance and connectivity maintenance of ASVs

The simulation results of formation control for Case C are shown in Fig.6-10. Fig.6 describes the formation trajectories of multiple ASVs with considering collision avoidance and connectivity maintenance performance. It can be seen that

the cooperative formation control for ASVs subject to unknown uncertainties and disturbances can be realized through the proposed distributed controller. Fig.7 shows the distance between all follower vehicles, from which it can be seen that all distances between any two inter-vehicles are greater than the minimum separation distance and less than the maximum separation distance at any time. It is inferred that the problems of collision avoidance and connectivity maintenance in Case A and B are effectively solved. Fig.8 and Fig.9 provide the tracking errors with respect to position and velocity. It further reveals that the proposed distributed controller can completely achieve the control objectives of formation generation, formation maintenance, collision avoidance and connectivity maintenance. Fig.10 shows the adaptive weight parameters of the neural network. It can be seen that the NN can approximate unknown uncertainties and disturbances by updating the adaptive weight parameters online. Compared with previous NN-based works, using NNs to approximate

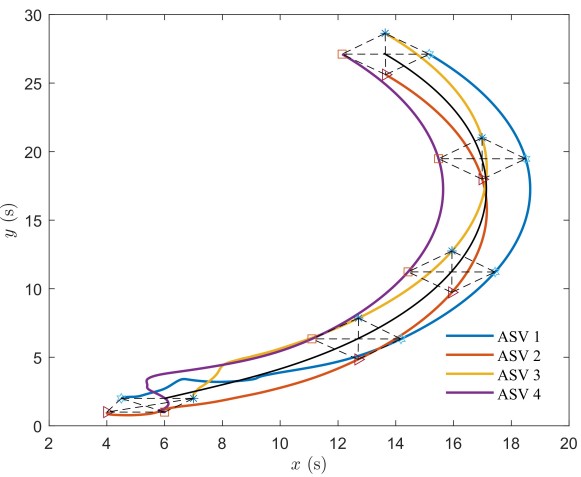

Fig. 6. The performance of formation control with collision avoidance and connectivity maintenance.

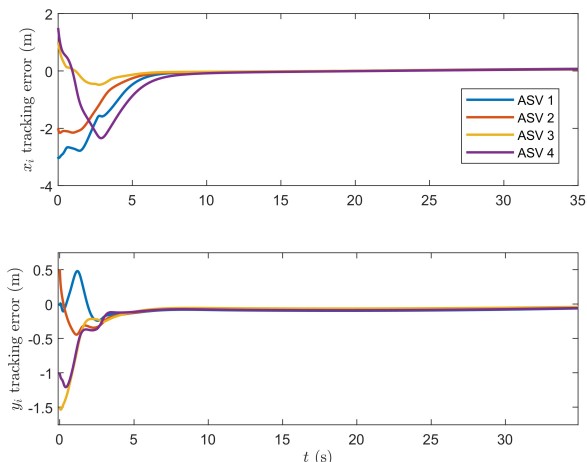

Fig. 8. The position tracking errors.

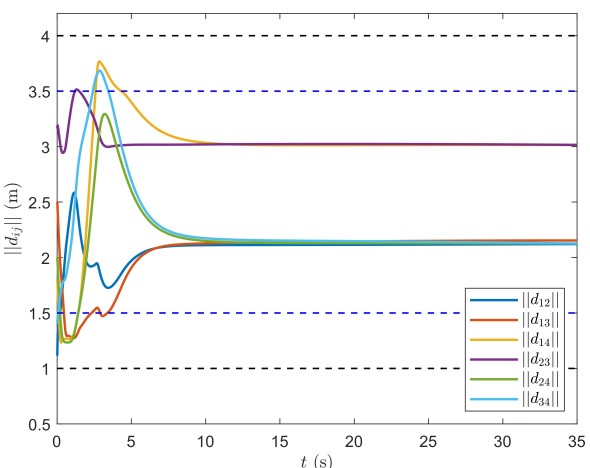

Fig. 7. The distance between all inter-vehicles.

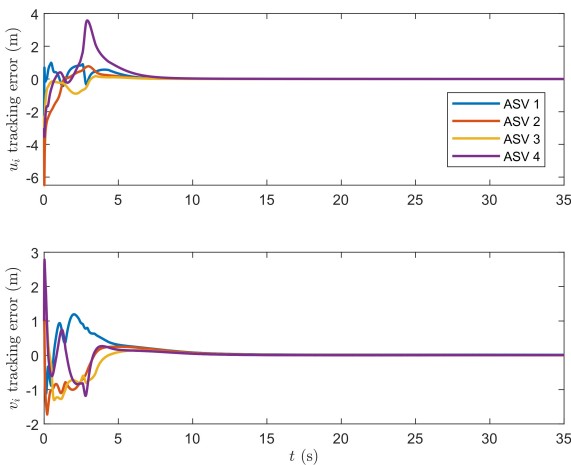

Fig. 9. The velocity tracking errors.

collective nonlinear dynamics requires at least three adaptive weight parameters to be updated for each ASV. In this paper, the proposed algorithm can greatly reduce the number of adaptive weight parameters to only one for each vehicle. Note that it means that only four adaptive weight parameters are required for the four ASVs in the simulation, rather than the twelve weight parameters of other NN-based algorithms.

## VI. Conclusion

This paper has investigated a distributed adaptive formation control with collision avoidance and connectivity maintenance strategy for multiple ASVs subject to unknown uncertainties and disturbances. Combining with APF methods into the leader-follower strategies, the distributed formation controller can achieve formation generation, formation maintenance, collision avoidance, and connectivity maintenance, simultaneously. NN technologies are employed to against the unknown

uncertainties and disturbances. The adaptive weight parameters are designed in scalar form, which can greatly reduce the number of weight parameters. Furthermore, three different simulation cases prove the effectiveness of the proposed distributed controller to against unknown uncertainties and disturbances and to achieve the four control objectives.

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
