# OpenReview forum: "Distributed Adaptive Formation Control with Collision Avoidance and Connectivity Maintenance of Multiple Autonomous Surface Vehicles"
_IEEE.org/ICIST/2024/Conference — IEEE ICIST 2024 Conference Submission_

### Official Review · Reviewer_YHcu · 2024-08-25
**Distributed Adaptive Formation Control with Collision Avoidance and Connectivity Maintenance of Multiple Autonomous Surface Vehicles**

**Rating:** 9
**Confidence:** 3

**Review:**

This paper is rich and engaging in content. The research focuses on distributed adaptive formation control for multiple autonomous surface vessels under uncertainty and disturbances. The artificial potential field method is employed to achieve the objectives of collision avoidance and connectivity maintenance in the formation.

---

### Official Review · Reviewer_8k8M · 2024-08-25
**minor repair**

**Rating:** 8
**Confidence:** 3

**Review:**

1. There are citation irregularities in this manuscript, for example: (i) If Lemmas 1 and 2 are not first introduced by the authors, it is recommended that the corresponding references be added.
2. The author should provide the learning effect of neural networks on uncertainty in the simulation section.
3. For the convenience of readers' understanding, it is suggested that the author add a flowchart of the control process

---

### Official Review · Reviewer_hiY6 · 2024-09-02
**This paper can be accepted.**

**Rating:** 7
**Confidence:** 5

**Review:**

This paper investigates a distributed adaptive formation control with collision avoidance and connectivity maintenance strategy for multiple autonomous surface vehicles. However in the reviewer's opinion, there are somecomments in the paper which should be addressed by the authors:
1. In addition to the contribution of the proposed work, it is suggested to discuss its main limitations/shortcomings.
2. Please add a structural flowchart of the paper's framework to enhance the readability of the paper.
3.The font in Figure 2 should be larger for better readability.
4.The English grammar and format of this manuscript could be further polished and checked carefully.

---

### Decision · Program_Chairs · 2024-09-06

Accept (Oral)